

# A retrospective observational study on maternal and neonatal outcomes of COVID-19: Does the mild SARS-CoV-2 infection affect the outcome?

Jing Li[1,2,*], Xiang Li[3,*], Peiying Ye[3], Yun You[3], Yu Wang[3], Jing Zhang[3], Weihua Zhao[3], Zhiying Yu[1], Runsi Yao[3] and Jie Tang[3]

[1] Department of Gynecology, Shenzhen Second People's Hospital, The First Affiliated Hospital of Shenzhen University, Shenzhen, Guangdong, China
[2] Guangdong Key Laboratory for Biomedical Measurements and Ultrasound Imaging, National-Regional Key Technology Engineering Laboratory for Medical Ultrasound, School of Biomedical Engineering, Shenzhen University Medical School, Shenzhen, Guangdong, China
[3] Department of Obstetrics, Shenzhen Second People's Hospital, The First Affiliated Hospital of Shenzhen University, Shenzhen, Guangdong, China
* These authors contributed equally to this work.

Corresponding authors
Runsi Yao, yaorunsi@163.com
Jie Tang,
tangjie907@email.szu.edu.cn

## ABSTRACT

**Background:** Currently, several SARS-CoV-2 variants, including Omicron, are still circulating globally. This underscores the necessity for a comprehensive understanding of their impact on obstetric and neonatal outcomes in pregnant women, even in cases of mild infection.

**Methods:** We conducted a retrospective, single-center observational study to investigate the association between gestational SARS-CoV-2 infection and maternal-fetal outcomes in the Chinese population. The study enrolled 311 pregnant patients with SARS-CoV-2 infection (exposure group) and 205 uninfected pregnant patients (control group). We scrutinized the hospital records to collect data on demographics, clinical characteristics, and maternal and neonatal outcomes for subsequently comparison.

**Results:** Similar characteristics were observed in both groups, including maternal age, height, BMI, gravidity, parity, and comorbidities ($p > 0.05$). A majority (97.4%) of pregnant women in the exposure group with COVID-19 experienced mild clinical symptoms, with fever (86.5%) and cough (74.3%) as the primary symptoms. The exposure group exhibited significantly higher incidences of cesarean section and fetal distress compared to the control group ($p < 0.05$). Furthermore, pregnant women in the exposure group showed reduced levels of hemoglobin and high-sensitivity C-reactive protein, while experiencing significantly increased levels of lymphocytes, prothrombin time, alanine aminotransferase, and aspartate aminotransferase ($p < 0.05$). Notably, recent SARS-CoV-2 infection prior to delivery appeared to have an adverse impact on liver function, blood and coagulation levels in pregnant women. When comparing the two groups, there were no significant differences in the postpartum hemorrhage rate, premature birth rate, birth weight, neonatal asphyxia rate, neonatal department transfer rate, and neonatal pneumonia incidence.

**Conclusions:** Our study suggests that mild COVID-19 infection during pregnancy does not have detrimental effects on maternal and neonatal outcomes. However, the increased risks of events such as fetal distress and cesarean section, coupled with potential alterations in physical function, reveal the consequences of SARS-CoV-2 infection during pregnancy, even in mild cases. These findings emphasize the importance of proactive management and monitoring of pregnant individuals with COVID-19.

## INTRODUCTION

The infection caused by the severe acute respiratory syndrome coronavirus 2 (SARS-CoV-2) infection, leading to the severe respiratory disease known as coronavirus disease 2019 (COVID-19), has presented unprecedented challenges on global human health since its emergence in December 2019 (*Kumar, Verma & Mysorekar, 2023*; *Zambrano et al., 2020*). The clinical manifestations and prognostic outcomes of SARS-CoV-2 infection display significant variation among different populations. This variation is influenced by a range of factors, including comorbidities, advanced age, pregnancy, immunosuppression, and malignancy (*Di Mascio et al., 2020*; *Dubey et al., 2022*; *Huang et al., 2020*). Pregnant women, in particular, are at an elevated risks of acquiring pathogen, experiencing COVID-19-related complications, and developing severe illness compared to their non-pregnant counterparts. This is due to dynamic changes in hormonal profiles, cardiovascular dynamics, coagulation cascades, and maternal immune adaptations that occur during gestation (*Chen et al., 2020*; *Yan et al., 2020*; *Zaigham & Andersson, 2020*). Furthermore, maternal infection may increase the likelihood of adverse outcomes in neonates through vertical transmission (*Adhikari et al., 2022*; *Kotlyar et al., 2021*). Recent investigations have reported that a significant proportion of infected pregnant women, ranging from approximately 79.8% to 94.6%, experience mild to moderate symptoms of COVID-19, while approximately 1.99% to 16.7% develop severe disease (*Arakaki et al., 2022*; *Kumari, Anand & Vidyarthi, 2022*; *Samadi et al., 2021*). Additionally, several studies have shown an increased hospitalization rate and a higher incidence of preterm births associated with COVID-19 in pregnant women compared to non-pregnant individuals (*Ellington et al., 2020*; *Sutton et al., 2020*).

The continued global transmission and devastating consequences of the severe acute respiratory syndrome coronavirus 2 (SARS-CoV-2) pandemic, exacerbated by the emergence of novel variants such as Omicron, persist as a profound threat to public health (*World Health Organization (WHO), 2023*). In the field of gynecology and obstetrics, the impact of COVID-19 on the well-being of pregnant women and their neonates remains a pressing concern, even in cases of mild infections. The scarcity of comprehensive information and evidence regarding the clinical manifestations and complications of COVID-19 during pregnancy represents a significant knowledge gap, partly attributable in part to the regional and racial disparities observed in previous epidemiological

investigations of COVID-19 (*Adhikari et al., 2022*; *Arakaki et al., 2022*; *Kumar, Verma & Mysorekar, 2023*; *Samadi et al., 2021*). Therefore, the objective of this study is to assess the clinical characteristics, as well as maternal and infant outcomes, of pregnancies affected by SARS-CoV-2 infection within the population of southern China. By undertaking this investigation, we aim to enhance the evolving guidelines and management strategies pertaining to pregnancies impacted by COVID-19, thereby contributing to the advancement of knowledge and the optimization of care in this crucial domain.

We hypothesized that severity of maternal COVID-19 infection would impact neonatal outcomes including preterm birth, neonatal birth weight, neonatal infection, and neonatal intensive care unit (NICU) admission in Chinese population.

## MATERIAL AND METHODS

### Research design

This retrospective, single-center, observational study was conducted at the First Affiliated Hospital of Shenzhen University, also known as Shenzhen Second People's Hospital. Initially, a total of 531 pregnant women who gave birth between November 01, 2022, and January 31, 2023, were included in the study. All these women had recorded SARS-CoV-2 PCR test results from nasopharyngeal swab samples taken during pregnancy and on admission, indicating either positive or negative results. From the initial pool, 15 pregnant women were excluded due to twin pregnancies (eight cases) or voluntary termination of pregnancy (seven cases). Ultimately, the study comprised 516 pregnant women, with 311 in the COVID-19 infected group (exposed group) and 205 in the uninfected group (control group). The medical records of these pregnant women were retrospectively reviewed, including information on clinical symptoms, medication usage, during pregnancy. Comparative analysis was performed on various data, including maternal demographics, medication usage during pregnancy, gestational age at COVID-19 diagnosis, antenatal diseases following COVID-19 infection, fetal growth restriction (FGR), fetal distress, intrapartum complications, gestational age at delivery (GA), mode of delivery, postpartum hemorrhage (PPH), birth weight, and admission to the maternal Intensive Care Unit (ICU) and Neonatal Intensive Care Unit (NICU). Pregnant women who tested positive for SARS-CoV-2 PCR but did not exhibit COVID-19 symptoms were classified as asymptomatic, while symptomatic cases were further categorized as mild, moderate, or severe based on the National Institutes of Health (NIH) COVID-19 classification guidelines (*NIH, 2023*). Pregnant women with confirmed COVID-19 were treated in designated isolation wards. Furthermore, data from pregnant women who delivered within 14 days of COVID-19 infection were compared with those who delivered >14 days after infection to assess the potential impact of acute COVID-19 infection on clinical characteristics and maternal and neonatal outcomes. The study protocol was approved by the Medical Ethics Review Committee of the First Affiliated Hospital of Shenzhen University (No. 2023-008-02PJ). All participants' identities were kept anonymous, and verbal informed consent was obtained.

## Statistical analysis

Statistical analyses were performed using SPSS Statistics software (version 25; SPSS Inc., Chicago, IL, USA). Continuous variables were expressed as mean ± standard deviation (SD), and categorical variables were presented as frequencies and percentages. Associations between categorical variables were assessed using the Chi-square test or Fisher's exact test, while the independent t-test or Mann-Whitney U test was used for continuous variables, as appropriate. Statistical significance was defined as a *p*-value of less than 0.05.

## RESULTS

### Demographic characteristics

A total of 531 pregnant women were included in the study and followed up. Among them, 516 met the inclusion criteria, constituting 311 pregnant women with SARS-CoV-2 infection (exposure group) and 205 uninfected pregnant women (control group). To mitigate potential confounding factors that could introduce bias, 15 cases were excluded: 8 involved twin pregnancies, and seven participants opted for pregnancy termination for specific reasons. Table 1 presents the demographic characteristics of the pregnant women in both groups.

The mean age of pregnant women in the exposure group was 31.63 ± 4.39, with an average BMI of 26.54 ± 3.80 kg/m². In contrast, the mean age of pregnant women in the control group was 32.04 ± 4.18, with an average BMI of 26.42 ± 3.72 kg/m². No significant differences were observed between the two groups in terms of age, height, BMI, parity, or comorbidities such as diabetes mellitus, hypertension, and hypothyroidism ($p > 0.05$) (Table 1).

According to the NIH classification (Table S1) (*NIH, 2023*), the majority of pregnancies in the exposed group were either asymptomatic or presented with mild infection (97.4%, $n = 303$). The most commonly observed symptoms were fever (86.5%, $n = 269$) and cough (74.3%, $n = 231$). Other common symptoms included sore throat (63.0%, $n = 196$), muscle aches (60.5%, $n = 188$), and fatigue (61.4%, $n = 191$) (Table 2). Additionally, five pregnant women (1.6%) exhibited moderate to severe manifestations, with X-ray imaging characteristics indicative of pneumonia caused by coronavirus infection. Another three pregnant women (1.0%) experienced complications of multiple organ failure and required ICU treatment.

### Impact of COVID-19 on obstetric and neonatal outcomes

The obstetric and neonatal outcomes of the two groups are illustrated in Tables 3 and 4. In comparison to the control group, the gestational age of the exposed group was slightly shorter (270.37 ± 12.99 *vs* 272.18 ± 12.86, $p = 0.121$), while the cesarean section rate (50.8% *vs* 31.7%) and fetal distress rate (14.5% *vs* 5.9%) were significantly higher ($p < 0.05$). Surprisingly, the uninfected pregnant women group exhibited a significantly higher incidence of premature rupture of membranes (PROM) and amniotic fluid contamination compared to the SARS-CoV-2 infected pregnant women ($p < 0.05$). Furthermore, no

**Table 1 Maternal characteristics between the exposure and control groups.**

| Category | COVID-19 group | Control group | $t$ | $\chi^2$ | $p$ |
|---|---|---|---|---|---|
| Age (y) | 31.63 ± 4.39 | 32.04 ± 4.18 | 1.045 | | 0.296 |
| Height (cm) | 159.16 ± 5.70 | 159.26 ± 5.38 | 0.198 | | 0.843 |
| BMI (kg/m$^2$) | 26.54 ± 3.80 | 26.42 ± 3.72 | −0.341 | | 0.733 |
| Parity | | | | 0.302 | 0.582 |
| Nullipara | 107 (52.2%) | 170 (54.7%) | | | |
| Multipara | 98 (47.8%) | 141 (45.3%) | | | |
| Co-morbidities | 152 (48.9%) | 88 (42.9%) | | 1.757 | 0.185 |
| GDM | 55 (15.6%) | 25 (10.4%) | | | |
| Hypertension | 15 (4.2%) | 6 (2.5%) | | | |
| Thyroid disease | 14 (4.0%) | 23 (9.6%) | | | |
| Autoimmune disease | 19 (5.4%) | 18 (7.5%) | | | |
| Thrombosis | 3 (0.8%) | 4 (1.7%) | | | |
| Hematological disease | 29 (8.2%) | 19 (7.9%) | | | |
| Liver/kidney diseases | 39 (11.0%) | 10 (4.2%) | | | |
| Other diseases | 20 (5.7%) | 18 (7.5%) | | | |

**Note:**
Abbreviations: BMI, body mass index; GDM, gestational diabetes mellitus. Values are given as number (percentage) or mean ± SD.

**Table 2 Symptoms and clinical classification of 311 pregnancies with COVID-19.**

| Symptoms/clinical classification | Number (percentage) |
|---|---|
| Fever | 269 (86.5%) |
| Cough | 231 (74.3%) |
| Sore throat | 196 (63.0%) |
| Muscle aches | 188 (60.5%) |
| Fatigue | 191 (61.4%) |
| Anosmia | 89 (28.6%) |
| Diarrhea | 15 (4.8%) |
| Nausea | 33 (10.6%) |
| Vomiting | 21 (6.8%) |
| anorexia | 127 (40.8%) |
| Shortness of breath | 49 (15.8%) |
| Eye discomfort | 22 (7.1%) |
| Positive chest CT scan | 8 (2.6%) |
| COVID-19 clinical classification | |
| Asymptomatic or Mild | 303 (97.4%) |
| Moderate | 2 (0.6%) |
| Severe | 3 (1.0%) |
| Critical | 3 (1.0%) |

**Table 3  Maternal outcomes among pregnant women with and without COVID-19.**

| Category | COVID-19 group | Control group | t | χ² | p |
|---|---|---|---|---|---|
| Gestational age (day) | 270.37 ± 12.99 | 272.18 ± 12.86 | 1.555 | | 0.121 |
| Premature delivery | 28 (9.0%) | 18 (8.8%) | | 0.008 | 0.931 |
| Cesarean section | 158 (50.8%) | 65 (31.7%) | | 18.827 | 0.001 |
| Fetal distress | 45 (14.5%) | 12 (5.9%) | | 9.334 | 0.002 |
| PROM | 44 (14.1%) | 43 (21.0%) | | 4.109 | 0.043 |
| Amniotic fluid Contamination | 44 (14.1%) | 73 (18.0%) | | 9.815 | 0.044 |
| Postpartum hemorrhage | 15 (4.9%) | 16 (7.8%) | | 1.867 | 0.172 |

**Table 4  Neonatal outcomes among pregnant women with and without COVID-19.**

| Category | COVID-19 group | Control group | t | χ² | p |
|---|---|---|---|---|---|
| Birth weight (kg) | 3.11 ± 4.90 | 3.07 ± 5.02 | −0.836 | | 0.404 |
| Neonatal asphyxia | 4 (1.3%) | 0 (0%) | | – | – |
| Neonatal transfer | 84 (27.0%) | 57 (27.8%) | | 0.039 | 0.843 |
| Neonatal pneumonia | 10 (3.1%) | 6 (2.9%) | | 0.034 | 0.853 |

statistically significant differences were observed in the rates of premature delivery and postpartum hemorrhage between the two groups ($p > 0.05$).

The average weight of neonates in the exposure group and control group was 3.11 ± 4.90 and 3.07 ± 5.02 kg, respectively. Twenty-seven percent ($n = 84$) of the neonates were admitted to the neonatology department, and 3.1% ($n = 10$) experienced pulmonary infection. No significant differences were observed in birth weight, transfer rate, or incidence of pneumonia between the two groups ($p > 0.05$). Additionally, four cases (1.3%) in the exposure group experienced neonatal asphyxia.

## Impact of COVID-19 on laboratory findings and drug usage among pregnant women

When comparing the exposed group to the control group, significantly lower levels of hemoglobin and CRP were observed in the exposed group, while levels of AST, ALT, lymphocytes, prothrombin time and the usage rate of antibiotics and antiviral drugs were higher ($p < 0.05$) (Table 5). Additionally, there were no significant differences in leukocytes, fibrinogen levels, usage rate of anticoagulant drugs, or incidence of thrombosis between the two groups ($p > 0.05$).

## Impact of recent SARS-CoV-2 infection on laboratory findings among pregnant women

In this cohort, all SARS-CoV-2 infections occurred during the third trimester of pregnancy. The time range from COVID-19 diagnosis to delivery was 0–70 days, based on which the cases were further divided into two subgroups: the recent infection group (≤14 days) and the non-recent infection group (>14 days). Compared to uninfected and

**Table 5 Laboratory findings and drug usage among pregnant women with and without COVID-19.**

| Category | COVID-19 group | Control group | t/Z | $\chi^2$ | p |
|---|---|---|---|---|---|
| Leucocytes ($\times10^9$/L) | 9.51 ± 2.92 | 9.65 ± 2.81 | 0.547* | | 0.566 |
| Hemoglobin (g/L) | 117.01 ± 15.19 | 120.79 ± 12.44 | 3.081* | | 0.002 |
| Lymphocytes ($\times10^9$/L) | 1.67 ± 1.84 | 1.66 ± 0.46 | −3.094# | | <0.001 |
| CRP (mg/L) | 18.73 ± 2.787 | 41.3 ± 2.382 | −2.662# | | 0.008 |
| ALT (U/L) | 22.82 ± 30.50 | 16.21 ± 15.67 | −4.107# | | <0.001 |
| AST (U/L) | 24.92 ± 15.49 | 20.91 ± 12.81 | −4.301# | | <0.001 |
| Prothrombin time (sec) | 14.11 ± 3.13 | 11.22 ± 0.08 | −2.904# | | 0.004 |
| Fibrinogen (g/L) | 4.47 ± 0.87 | 4.45 ± 0.74 | −0.112# | | 0.911 |
| Thrombosis | 3 (1.0%) | 3 (1.5%) | | 0.262 | 0.686 |
| Antibiotic drugs usage | 186 (59.8%) | 99 (48.3%) | | 6.625 | 0.010 |
| Anticoagulant drug usage | 92 (29.6%) | 58 (28.3%) | | 0.100 | 0.752 |
| Antiviral drugs usage | 3 (1.0%) | 0 (0%) | | 516.0 | <0.001 |

**Notes:**
Abbreviations: ALT, alanine aminotransferase; AST, aspartate aminotransferase; CRP, C-reactive protein.
* t test.
# Mann-Whitney U test.

**Table 6 Laboratory findings among pregnant women with and without recent infection.**

| Category | ≤14 days | >14 days | Uninfected | $\chi^2$ | p |
|---|---|---|---|---|---|
| Hyperleukocytosis (>10 × $10^9$/L) | 28 (26.2%) | 86 (42.2%) | 75 (36.6%) | 7.730 | 0.021 |
| Lymphopenia (<1.1 × $10^9$/L) | 55 (51.4%) | 28 (13.7%) | 20 (9.8%) | 84.531 | <0.001 |
| Hemoglobin (g/L) | 25 (23.4%) | 57 (27.9%) | 33 (16.1%) | 8.372 | 0.015 |
| CRP (mg/L) | 47 (43.0%) | 22 (21.2%) | 29 (14.1%) | 55.296 | <0.001 |
| ALT (U/L) | 17 (15.9%) | 17 (8.3%) | 10 (4.9%) | 10.942 | 0.004 |
| AST (U/L) | 21 (19.6%) | 19 (9.3%) | 14 (6.8%) | 12.765 | 0.002 |

non-recently infected pregnant women, the proportion of lymphocyte decrease, as well as levels of ALT, AST, and CRP, were significantly higher in recently infected pregnant women ($p < 0.05$) (Table 6). In contrast, no significant differences were observed in the aforementioned indicators between uninfected and non-recently infected pregnant women.

# DISCUSSION

While SARS-CoV-2 is no longer considered a global public health emergency as of January 2023, it continues to spread worldwide, lresulting in a growing number of pregnant women with COVID-19 (*World Health Organization (WHO), 2023*). While studies from different countries have explored the impact of COVID-19 infection on maternal and neonatal outcomes in various ethnicities, research specific to the Chinese population is currently lacking. In this retrospective COVID-19 obstetric cohort study, we found that mild SARS-CoV-2 infection during pregnancy did not adversely affect maternal and neonatal

outcomes. However, it was associated with an increased risk of certain events, such as fetal distress and cesarean section.

In this study, all SARS-CoV-2 infections occurred during the third trimester of pregnancy, with an average gestational age of 246.7 ± 21.56 days at the time of SARS-CoV-2 infection. These findings are consistent with other studies conducted during the same period (*Antoun et al., 2020*; *Kumari, Anand & Vidyarthi, 2022*; *Santos et al., 2022*; *Yan et al., 2020*), highlighting the importance of continued prophylaxis throughout pregnancy. Approximately 15.6% of pregnant women with COVID-19 were diagnosed with gestational diabetes mellitus (GDM), compared to 10.4% in the control group. Similarly, several studies have also reported an increased risk of GDM among pregnant women with COVID-19 (*Ozbasli et al., 2023*; *Wei et al., 2021*). While a previous study by *Baracy et al. (2021)* reported an increased risk for gestational hypertension in COVID-19 pregnancies, only 4.2% of SARS-CoV-2-infected pregnant women in our study experienced gestational hypertension, and no significant association was observed between the maternal SARS-CoV-2-infection and gestational hypertension. This finding confirms observations made by *Chornock et al. (2021)*, *Wetcher et al. (2023)*. The pregnant women included in this cohort were infected with SARS-CoV-2 during the period when the Omicron variant was dominant. The majority of pregnant women with COVID-19 were either asymptomatic or experienced mild symptoms (97.4%), while only eight cases (2.6%) were classified as moderate to severe. Consistently, multiple studies during the same period reported that mild cases accounted for 79.8% to 94.6% among pregnant women with SARS-CoV-2 infection (*Arakaki et al., 2022*; *Kumari, Anand & Vidyarthi, 2022*; *Samadi et al., 2021*). The most common symptoms observed in SARS-CoV-2-infected pregnant women were influenza-like symptoms (including fever, cough, and sore throat), myalgia, and fatigue, which is consistent with observations made by *Samadi et al. (2021)*. An earlier meta-analysis study reported that 16.9% and 27.9% of SARS-CoV-2-infected pregnant women experienced ageusia and anosmia, respectively (*Figueiro-Filho, Yudin & Farine, 2020*). Similarly, our cohort of pregnant women cohort also observed a slightly higher proportion of anosmia (28.6%).

Previous sporadic studies have reported alterations in lymphocytes, C-reactive protein (CRP), and liver enzymes in individuals infected with COVID-19 (*Abedzadeh-Kalahroudi et al., 2021*; *Choudhary, Singh & Bharadwaj, 2022*; *Muhidin, Behboodi Moghadam & Vizheh, 2020*). Consistent with these findings, our study observed a significant decrease in hemoglobin and high-sensitivity CRP, as well as an increase in lymphocytes, prothrombin time, alanine aminotransferase, and aspartate aminotransferase levels in pregnant women with COVID-19 compared to the control group. These changes were particularly pronounced in pregnant women who gave birth within 14 days of SARS-CoV-2 infection, as opposed to non-infected pregnant women or those who delivered after 14 days of infection. These results highlight the adverse effects of SARS-CoV-2 infection on liver function, blood and coagulation levels in pregnant women, especially in cases of recent infection. Elevated levels of inflammatory factors in COVID-19 patients can activate the extrinsic coagulation pathway and inhibit the fibrinolytic system. Additionally, the activation of thrombin further promotes excessive production of inflammatory cytokines

(*Asakura & Ogawa, 2021*; *Zhang et al., 2021*). The hypercoagulable state during pregnancy normally serves as a physiological change to prevent postpartum hemorrhage (*Soma-Pillay et al., 2016*). However, SARS-CoV-2 infection may exacerbate the hypercoagulable state in pregnant women. Previous studies have indicated an increased incidence of thrombosis in pregnant women with COVID-19 (*Metz et al., 2022*; *Nasir & Ahmad, 2021*), although we did not observe such changes in our study. This may be partially attributed to the relatively slow process of thrombosis and the prophylactic use of anticoagulants. Both the exposure and control groups had a high usage rate of anticoagulant drugs, with rates of 29.6% and 28.3%, respectively.

In this study, the COVID-19 infected pregnant women had a slightly lower gestational age at delivery compared to the uninfected pregnant women, and there was no significant difference in the incidence of preterm delivery between the two groups. Previous evidence has suggested a higher risk of preterm delivery among pregnant women with COVID-19 (*Panagiotakopoulos et al., 2020*; *Villar et al., 2021*; *Wei et al., 2021*; *Woodworth et al., 2020*). However, these studies mainly focused on severe late-pregnancy infections in hospitalized cases, with patients who had early infection and milder symptoms being underrepresented. In our study, the rate of preterm delivery among pregnant women with COVID-19 was 9.0%, slightly lower than the previously reported 10–11% during the Omicron period (*Deng et al., 2022*; *Favre et al., 2023*). This finding is reasonable, possibly because all SARS-CoV-2 infections occurred during the third trimester of pregnancy. Additionally, the majority of infections in pregnant women during this study were mild, which might not significantly impact the preterm birth rate compared to uninfected pregnant women. The cesarean section rate was significantly higher in pregnancies with COVID-19 compared to the control group (50.8% *vs* 31.7%), which aligns with previous observations (*Abedzadeh-Kalahroudi et al., 2021*; *Gurol-Urganci et al., 2021*; *Yan et al., 2020*). It is worth noting that the rate of fetal distress was significantly higher in pregnancies with COVID-19 than in the control group, and there was also a more noticeable increase in infection-related indicators and liver enzymes. These factors may influence the delivery decisions of obstetricians and pregnant women, resulting in a higher rate of cesarean section. Furthermore, the incidence of premature rupture of membranes and amniotic fluid contamination was higher in the control group than in pregnant women with COVID-19 ($p < 0.05$). This finding is not contradictory, as a prolonged gestational age and a higher rate of vaginal delivery rate were also observed among the SARS-CoV-2 uninfected pregnant women, which may contribute to the increased risk of premature rupture of membranes and amniotic fluid contamination.

Early evidence has indicated high rates of Neonatal Intensive Care Unit (NICU) admission, severe perinatal morbidity, and mortality among COVID-19-positive pregnant women (*Brandt et al., 2021*; *Wei et al., 2021*). However, in this study, no significant differences were observed between the two groups in terms of birth weight, pneumonia incidence rate, and transfer rate ($p > 0.05$). This finding may be attributed to the weakened pathogenicity of Omicron variants and the widespread coronavirus vaccination, both of which enhance the protection of pregnant women (*Auger & Healy-Profitós, 2022*; *Male, 2022*; *Onyinyechi Chionuma et al., 2022*). It is important to note that this study did not

investigate evidence of neonatal SARS-CoV-2 infection, thus no conclusion could be drawn regarding the existence of vertical transmission. Limited observations from newborns infected with Omicron variants have suggested that neonatal infection is usually asymptomatic or mild, with no specific clinical manifestations and a generally good short-term prognosis (*Auger & Healy-Profitós, 2022*; *Mndala et al., 2022*; *Stock et al., 2022*). However, further studies are needed to investigate the long-term prognosis of neonatal SARS-CoV-2 infection. Furthermore, this study did not include multiple pregnancies, and pregnant women infected before or during the first and second trimesters were not analyzed due to the absence of such cases during that period. Therefore, we were unable to explore the potential impact of COVID-19 throughout the entire pregnancy or its effects on women with multiple pregnancies. Future research should aim to address these important areas of investigation.

## CONCLUSIONS

The current study findings suggest that mild SARS-CoV-2 infection during late pregnancy is not appear to be associated with adverse obstetric and neonatal outcomes in the Chinese population. However, it does indicate a potential increased risk for events such as fetal distress and cesarean delivery. Additionally, this study underscores the potential adverse effects of recent SARS-CoV-2 infection on liver function, blood and coagulation levels in pregnant women. These findings offer valuable insights for obstetric counseling and recommendations for pregnant women with COVID-19. Future studies should consider these results when investigating the impact of infection on maternal and neonatal health, as well as the potential risk of SARS-CoV-2 infection in newborns.

### Funding

This study was supported by the Shenzhen Second People's Hospital Clinical Research Fund of Shenzhen High-level Hospital Construction Project (Grant No. 2023YJLCYJ015). The funders had no role in study design, data collection and analysis, decision to publish, or preparation of the manuscript.

### Grant Disclosures

The following grant information was disclosed by the authors:
Shenzhen Second People's Hospital Clinical Research Fund of Shenzhen High-level Hospital Construction Project: 2023YJLCYJ015.

### Competing Interests

The authors declare that they have no competing interests.

### Author Contributions

- Jing Li conceived and designed the experiments, analyzed the data, prepared figures and/or tables, authored or reviewed drafts of the article, and approved the final draft.

- Xiang Li performed the experiments, prepared figures and/or tables, and approved the final draft.
- Peiying Ye performed the experiments, prepared figures and/or tables, and approved the final draft.
- Yun You performed the experiments, prepared figures and/or tables, and approved the final draft.
- Yu Wang performed the experiments, prepared figures and/or tables, and approved the final draft.
- Jing Zhang performed the experiments, prepared figures and/or tables, and approved the final draft.
- Weihua Zhao analyzed the data, authored or reviewed drafts of the article, and approved the final draft.
- Zhiying Yu analyzed the data, prepared figures and/or tables, and approved the final draft.
- Runsi Yao conceived and designed the experiments, analyzed the data, authored or reviewed drafts of the article, and approved the final draft.
- Jie Tang conceived and designed the experiments, analyzed the data, authored or reviewed drafts of the article, and approved the final draft.

### Human Ethics

The following information was supplied relating to ethical approvals (*i.e.*, approving body and any reference numbers):

The Medical Ethics Review Committee of the First Affiliated Hospital of Shenzhen University.

### Data Availability

The raw data is available in the Supplemental File.

### Supplemental Information

Supplemental information for this article can be found online at http://dx.doi.org/10.7717/peerj.16651#supplemental-information.

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
