# Peer review of "A retrospective observational study on maternal and neonatal outcomes of COVID-19: Does the mild SARS-CoV-2 infection affect the outcome?"

_PeerJ, doi:10.7717/peerj.16651_

## Round 0.1 · original submission · Major Revisions

The reviewers have suggested that your manuscript requires a great detail of information. I agree with the reviewers that more work needs to be carried out. Also, check the manuscript for grammatical errors, the language needs further improvement for readability.

**Language Note:** The Academic Editor has identified that the English language must be improved. PeerJ can provide language editing services - please contact us at [email protected] for pricing (be sure to provide your manuscript number and title). Alternatively, you should make your own arrangements to improve the language quality and provide details in your response letter. – PeerJ Staff

·

Basic reporting

Li et al manuscript was well-written, clear, and followed the standard for reporting scientific findings. However, the English language should be improved to ensure a clear understanding of the article, especially the abstract.

Experimental design

Li et al manuscript included a few aspects of the standard for designing a study, however, the research questions and hypothesis of the study are missing. Readers can easily understand, but authors are encouraged to provide these details

Validity of the findings

The Data presented are sound, and the statistical approach is okay. Supplementary data on the method of pneumonia scoring (line 153) should be provided. The authors over-emphasized the immune outcome result, the manuscript lack data to support this. Authors should also double-check for wrong reporting of the findings. For clarity see the additional comment tab.

Conclusions are well stated but need to be linked to the research question/hypothesis of the study

Additional comments

General comment
Main concern
• The author stated in the result section of the abstract that SARS-CoV-2 adversely impacts liver function, immune defense, and inflammation levels in pregnant women. This manuscript has no sufficient data to support the author’s claims on the impact of SARS-CoV-2 on immune defence and inflammation. For instance, cytokine data, and differential leukocytes data are missing, which are the basics. I suggest the authors state their findings on the immune outcome as it appears without overestimating. Examples: doi: 10.1038/s41467-021-27745-z; doi: 10.3390/children10050771
• Line 152 – 154, the percentage for the x-ray data (5(1.5%, and 3(0.9%) did not correlate with the data as presented in Table 2. Authors are advised to submit supplementary data on the X-ray scan and scoring for pneumonia as claimed in line 153.
• Line 160: Authors should revise this statement with data presented in Table 3. There was no significant difference between groups as presented in Table 3
• Line 167; authors should revise the sentence and include the birth weight of the control group
• Line 176 - 178: Part of this statement is wrong, which makes it difficult to follow through. Refer to Table 5 for CRP interpretation. There was a significant in CRP level in the infected group compared to the control (refer to Table 5).
• No data on procalcitonin, this should not be mentioned in the result or abstract section
• Line 226: advice to remove procalcitonin, as there was no data on this in the authors’ study

Minor concern
• Line 104, authors should provide information on COVID-19 vaccination during pregnancy in the data or remove the phrase.
• Line 176: there should be a comma between lymphocyte and hemoglobin. There is multiple repetitions of the word “lymphocytes” in the sentence

Reviewer 2 ·

Basic reporting

In this study by Li J. et al., authors aimed to examine the association of maternal and neonatal outcomes in mild SARS-Cov2 infection patients using a single center observational study design.
However, there are multiple recent studies already had been published on same subject shwing non-significant or minute impact of SARS-Cov2 infection during pregnancy. Below are the published studies-
1.Ohttps://doi.org/10.1016/S2214-109X(22)00359-X
2. doi:10.1001/jama.2021.5775
3. https://doi.org/10.1177/10547738211064027
4. DOI: 10.7759/cureus.13184
5. DOI: 10.4103/jfmpc.jfmpc_1321_21

Therefore, this manuscript does not contribute any significant information in already existed knowledge pool.

Experimental design

No comment

Validity of the findings

No comment

Reviewer 3 ·

Basic reporting

Language: clear, professional, unambiguous

Experimental design

Research methodology: Observational, retrospective. Well defined. With the structure of Theoretical Framework, Question, Material and methods, results, conclusions, and discussion based on the research question

Validity of the findings

The results are robust, with little population loss due to controlled exclusion.
The conclusion is well-founded, since it is observational, potential sampling biases are clarified.

Additional comments

I consider it a well-planned and developed paper.
It sheds new knowledge on the question at hand: "COVID-19 in pregnant women"

---

## Round 0.2 · Minor Revisions

Please check through for conciseness and clarity.
E.g. what "slightly shorter average" means in statistics, especially if it is not statistically significant.

Inconsistent findings are observed between the data and statements regarding prothrombin time and lymphocyte count.

·

Basic reporting

Li et al manuscript is consistent with existing literature. However, I still recommend that the English language should be improved to ensure clear understanding.

Furthermore, the author should revise the abstract section of this manuscript.

Experimental design

This section had been improved, and the authors had addressed my concerns

Validity of the findings

The authors had made general improvement on the manuscript, however, the abstract should be revised carefully. There are misinterpretation of data.

Additional comments

Major concern
Table 3, Line 263 – 265. From the data, the cohort in this study only include pregnant women who had delivered term babies (270.3 days, approximately 38.5 weeks). Gestational age for preterm birth according to WHO are babies born before 37 weeks of gestational age i.e., 28 – 34 weeks and this depends on the stage of preterm. This explains why there is no significant differences in GA between groups, the data variation is close. Authors should re-examine if proper comparisons where effectively done.

Minor concern

Line 36 (Abstract section): The exposed group exhibited gestational weeks….. This statement is wrong as there is no significant difference p = 0.121.
Line 38 – 40. Li et al stated that pregnant women in the exposure group showed reduced levels of hemoglobin, lymphocytes, fibrinogen, and prothrombin time. The data presented in Table 5 contradict this statement. For instance data shows increased level of lymphocytes, fibrinogen and prothrombin time.
Line 169. There is noting as slightly shorter average in statistics, if it is not significant, this should be clearly stated.
Line 186 – 188. Data contracts this statement. There is an increase in prothrombin time, and lymphocytes count.
Line 223 – 226 . Authors should check this sentence, some words are missing, which makes it difficult to understand.

Reviewer 3 ·

Basic reporting

The article performs satisfactorily across all evaluation criteria. It employs clear and professional English, making the content accessible and comprehensible. The extensive use of literature references and the provision of adequate field background/context enhance the article's credibility and depth. The professional article structure, accompanied by well-constructed figures and tables, facilitates a structured and informative reading experience. The sharing of raw data promotes transparency, and the article maintains its self-contained nature while consistently presenting results that align with the hypotheses. Overall, the article is in good standing with respect to these evaluation criteria.

Experimental design

The research aligns well with the established Aims and Scope of the journal, effectively delving into original primary research that addresses a critical knowledge gap in the field. The research question is thoughtfully defined, demonstrating relevance and significance within the broader context. The investigation maintains a rigorous technical and ethical standard, ensuring methodological integrity throughout. The methods are adequately detailed, allowing for potential replication, and promoting transparency and scientific rigor.

Validity of the findings

The research demonstrates a satisfactory level of performance across the evaluated criteria. While the assessment of impact and novelty could be more explicit and the rationale for encouraging meaningful replication clarified, it is nonetheless positive to see the provision of all robust, statistically sound, and well-controlled underlying data. The conclusions effectively align with the original research question and are appropriately constrained to supporting the presented results.

Additional comments

I consider the authors have answered the reviewers' questions in an acceptable manner.
It is appreciated that they know the topic they have investigated, and have worked on it with the best analysis tools.

---

## Round 0.3 · accepted · Accept

All reviewer's comments have been adequately addressed.